# Selenium–Fascinating Microelement, Properties and Sources in Food

**DOI:** 10.3390/molecules24071298

**Published:** 2019-04-03

**Authors:** Marek Kieliszek

**Affiliations:** Faculty of Food Sciences, Department of Biotechnology, Microbiology and Food Evaluation, Warsaw University of Life Sciences-SGGW, Nowoursynowska 159 C, 02-776 Warsaw, Poland; marek_kieliszek@sggw.pl or marek-kieliszek@wp.pl; Tel.: +48-22-593-7657; Fax: +48-22-593-7681

**Keywords:** selenium, food, supplementation, bioavailability, human health

## Abstract

Selenium is a micronutrient that is essential for the proper functioning of all organisms. Studies on the functions of selenium are rapidly developing. This element is a cofactor of many enzymes, for example, glutathione peroxidase or thioredoxin reductase. Insufficient supplementation of this element results in the increased risk of developing many chronic degenerative diseases. Selenium is important for the protection against oxidative stress, demonstrating the highest activity as a free radical scavenger and anti-cancer agent. In food, it is present in organic forms, as exemplified by selenomethionine and selenocysteine. In dietary supplementation, the inorganic forms of selenium (selenite and selenate) are used. Organic compounds are more easily absorbed by human organisms in comparison with inorganic compounds. Currently, selenium is considered an essential trace element of fundamental importance for human health. Extreme selenium deficiencies are widespread among people all over the world. Therefore, it is essential to supplement the deficiency of this micronutrient with selenium-enriched food or yeast cell biomass in the diet.

## 1. Introduction

Selenium was discovered in 1817 in Gripsholm, a Swedish city, by a Swedish chemist Jacob Berzelius [1], who was working in a chemical factory producing, among others, sulfuric acid and nitric acid. One of the raw materials that had been used in the production process was pyrite (iron sulfide), which was obtained from a mine in Falun. It was observed that, when ores from Falun were used in lead chambers of installations, reddish sediment was obtained. Initially, it was thought that arsenic compound was responsible for that characteristic sediment; hence, because of fear of its harmful effects, pyrite ores from Falun were avoided from being processed. The phenomenon was, however, regarded as curious and worth further assessment.

The results of sample analysis indicated a likely presence of tellurium; however, Berzelius questioned this result. At the beginning of 1818, Berzelius repeated the experiments in a laboratory in Stockholm and found that the sediment investigated contains a new, previously undiscovered element, with properties similar to sulfur [2]. This substance was called selenium, from the Greek word “selene”, which means moon [3].

Increased interest in the biological role of selenium was observed in the 1950s, when it was discovered that this element exerts toxic effects. Increased accumulation of this element led to dystrophy of cardiac muscle or acute hepatic necrosis [4]. In 1973, the biochemical role of this element was explored, and it was found that selenium belongs to a part of the active center of glutathione peroxidase. After 17 years, it was observed that other enzymes also include this element in their active centers, for example, selenocysteine is a part of the active center of the iodothyronine deiodinase. Studies conducted on the identification of selenoenzymes and selenoproteins initiated intensive studies on the role of this element in human and animal organisms [1].

## 2. Physicochemical Properties of Selenium

Selenium is found in group 16 of the periodic table, which also includes oxygen, sulfur, polonium, tellurium, and livermorium [5]. This group is called oxoacids. Chemical properties of the elements in this group vary significantly with the increase in the atomic mass of the elements (Table 1). Oxygen and sulfur are typical nonmetallic elements, selenium and tellurium are semimetals exhibiting properties of transition semimetals, and polonium has metallic characteristics. Selenium in the environment is found in elemental state (Se^0^), in the form of selenides (Se^2−^), selenates (SeO_4_^2−^), or selenites (SeO_3_^2−^) [6]. This element is characterized by an ease of transition to adjacent oxidation states. These transformations are influenced by several factors such as pH, concentration of free oxygen, redox potential, and humidity. Anaerobic conditions and acidic environment favor the formation of selenium compounds in lower oxidation states. Under aerobic conditions and at alkaline pH, higher oxidation states of this element are dominant [7].

Selenium in a free state can be found in five allotropic forms, two of which are amorphous, and the remaining three are crystalline [8]. It can form molecules of ring structure, consisting of eight atoms and chain molecules that are characterized by a considerable length. Ring-shaped molecules Se8 are unstable and possess crystalline forms α and β characterized by a red color. These are formed as a result of transformation of red elemental selenium, which is obtained by the condensation of selenium vapor. Chain molecules are present in the molten selenium of high viscosity. Sudden cooling of such alloy leads to the formation of gray amorphous selenium also known as vitreous selenium [9]. Chain particles can also be found in gray selenium, also known as metallic selenium, which is the most stable variant of this element. This variant is obtained by heating other selenium varieties to a temperature of 470 K. It is characterized by a very low electrical conductivity in the dark, which increases significantly as a result of radiation. Most isotopic variations of selenium are stable, apart from ^75^Se radioactive isotope emitting β and γ radiations. In nature, the most widespread isotope of the element is ^80^Se [10].

## 3. Occurrence of Selenium in the Environment

Selenium is a commonly occurring element in nature. It can be found in the atmosphere, lithosphere, biosphere, and hydrosphere of the Earth [11]. This element circulating in the environment initiates the process of weathering in rocks. Selenium is emitted into the atmosphere through volcanic gasses. Biomethylation of this element by microorganisms, decomposition of organic matter rich in this element, and so on contribute the most to the constant enrichment of the atmosphere with selenium. In these processes, volatile selenium compounds such as dimethylselenium (DMSe), hydrogen selenide (H_2_Se), and selenium oxide (SeO_2_) are produced. The average selenium content in the arable layer of soil varies from 0.33 to 2 mg/kg on a global scale [12]. Soils that have arisen from parent rocks rich in selenium such as sandstones and limestone have been reported to have selenium in large content [7,13]. For example, in the area of Olkiluoto Nuclear Power Plant (Japan) in the organic part of the soil (humus), selenium is present at a level of 34 mg/kg. In mineral soil, regardless of its depth, the content level of this element fluctuates ~14 mg/kg [14,15].

In water, selenium is present in trace quantities, and mainly in the form of selenates and selenites [16]. The amount of selenium in groundwater is much higher than that in seawater [13] because of selenium elution from the parent rocks and excessive fertilization of soils with mixtures rich in selenium compounds [17]. The selenium content determined in groundwater in Poznań (Poland) is 0.17–0.44 µg/L [18]. According to the recommendations of the World Health Organization (WHO), the acceptable amount of selenium in drinking water is 10 μg/L [19].

Selenium can be found in many minerals such as berzelianite (Cu_2_Se), klaustalite (PbSe), and naumanite (Ag_2_Se) [16]. It penetrates the soil as a result of anthropogenic activity through the combustion of coal and lignite, crude oil, and the use of agrotechnical processes—fertilization or liming [14]. These treatments are being done for example in Hubei Province, an area of Enshi in China, where the selenium content in coal oscillates at the level of 6–8.4 g/kg [17]. However, the greater part of China surface exhibits a very low content of this element in the soil (<0.1 mg/kg).

In the basin of Apure river in Venezuela, which is a tributary at the left of the Orinoco River, the amount of selenium was observed at the level of 2.9 mg/kg. Soils rich in selenium are found in the United States (e.g., Dakota, Wyoming, and Kansas states), Russia, and several regions of China [7]. Soils rich in this element occur mainly in North America, Canada, Australia, Ireland, while selenium-deficient areas include New Zealand and a wide part of Europe. On the Polish territory, selenium is found at a low level in the environment (selenium deficiency occurs in >70% of the country areas) [20].

## 4. Selenium in Human Organism

Selenium is an essential bioelement that is necessary for the functioning of all organisms. The amount of this element present in nature and in the human organism is very diverse depending on the geographic region and diet. An optimal daily dose of this element is established at 55 µg [21] and affects the normal course of biochemical and physiological processes [22,23]. Selenium is present in the human organism in trace quantities. Serum selenium levels may differ among populations, depending on a number of factors, including, but not limited to, concentration of selenium in food. The concentration of this element in adult human blood serum depends on a person’s age [24]

The total amount of selenium in a human organism is ~3–20 mg. Skeletal muscles of the body are main organs containing ~46.9% of the total content of this element in humans, whereas kidneys contain only 4% of selenium [25]. The most commonly used indicator of “selenium status” is the determination of its concentration in serum, which is estimated at 60–120 ng/mL [26]. The maximum selenium concentration is achieved in adulthood. The content of this element in serum is progressively decreasing in individuals >60 years of age [27]. In the human body, deficiency of this element is observed when its amount in plasma is lower than 85 µg/L [28]. Low selenium concentration in plasma is associated with 4- to 5-fold increased risk of prostate cancer [29]. In the plasma of the residents of central Poland, selenium content is relatively low and is estimated to be ~50–55 µg/L [28].

For a long time, selenium was considered a toxic element. Poisoning with this element led to the development of severe anemia, bone stiffness, hair loss, and blindness [7]. These symptoms have been observed in humans and animals in areas where the content of this element in the soil was ~1000 times greater in comparison with soils with an average amount of selenium in the other regions of the world [25]. Selenium can also enter the body by inhalation; hence, its maximum concentration in the air should not exceed 0.2 mg/m^3^ [30]. It should be emphasized that either too high level of selenium or its deficiency is harmful to human health. The difference between a dose necessary for the proper functioning of the organism and a harmful dose is small [1,31]. The recommended daily dose of selenium is different depending on the geographical area. The World Health Organization (WHO) recommends a daily dose of selenium at a level of 55 µg for adults [32,33,34,35]. A daily dose of 400 µg is considered harmless [1]. Food and Nutrition Board (FNB) in the US has acknowledged that the amount of selenium needed changes with age and accounts for 40–70 µg for men and 45–55 µg for women [1,7,27]. For children, the recommended daily dose should be 25 µg. In Great Britain, the recommended dose of selenium is higher in comparison with other parts of Europe, accounting for 60 µg for women and 75 µg for men (Table 2).

Daily doses of this element in the range of 100–200 µg lead to the reduction of genetic damages [1]. In addition, it is considered that selenium may be an important factor in the prevention of cancer development [31,36,37]. Selenium consumption at the level of 200 µg per day in the form of Na_2_SeO_3_ leads to the increased formation of cytotoxic T cells and natural killer (NK) cells. The results of studies conducted in the UK have shown that consumption of 100 µg of selenium a day relieves the symptoms of depression and anxiety [28]. The dose that does not show any adverse effect for adults is estimated at 800 µg Se/day, whereas a dose that causes the onset of toxicity ranges from 1540 to 1600 µg Se/day. The risk of symptom occurrence varies in every individual for a particular organism [38]. The content of selenium in food products in a given geographical region is proportional to its amount in the soil present in that region. Therefore, the amount of this element in food differs for similar products from particular regions of the world. Therefore, the average supply of selenium in the diet depends on the geographical area. In the United States, it slightly exceeds 90 µg per day [31], whereas in Venezuela, it is estimated at 326 µg [1]. A daily intake of selenium by inhabitants living in the central Poland is ~30–40 µg [28]. In Finland, where once a low supply of this element was reported (at the level of 30 µg per day), selenium compounds were added to agricultural fertilizers, which then increased the supply of selenium in the diet of Finns to 125 µg per day [1,7]. In some European countries, the intake of this element is below the recommended value (~30 µg/day) [38,39].

## 5. Effect of Deficiency and Excess of Selenium on Human Health

Prolonged selenium deficiency in human organism leads to serious diseases. Deficiency of this element adversely affects the functioning of the cardiovascular system and can be a direct cause of myocardial infarction [13]. It is associated with endemic diseases: Keshan and Kashin-Beck. These diseases were identified for the first time in women of childbearing age and children in the area of China where very low amount of selenium was found in soil and crops [40]. During Keshan disease, degeneration of the heart muscle is observed. In the case of Kashin-Beck, osteoarthritis is reported leading to the degeneration of cartilage in the joints of arms or legs [28]. As a result of epidemiological studies, it was concluded that moderate deficiency of selenium in daily diet affects the development of diseases resulting from reduced immunity [41]. Selenium deficiency in daily diet can adversely affect the functioning of the nervous system [42]. Among individuals with selenium deficiency, development of depression, or intensification of anxiety is observed; Alzheimer’s disease is also developed [43]. This element is considered to be crucial in reducing the virulence of HIV and in decreasing the progression to full-blown AIDS [44]. Selenium deficiency in pregnant women negatively affects the development of the embryo [13,45]. Excess of selenium can be toxic to the organism. Acute selenium poisoning is rarely observed. The accurate determination of harmful doses of selenium is difficult because of the occurrence of various chemical forms of this element. A toxic effect on the organism can be exerted by both organic and inorganic forms of selenium [46]. Toxicity of selenium (depends on the dose) is associated with competitive inhibition between selenium and sulfur, leading to the onset of sulfur metabolism (transformation) [47]. Selenium may substitute sulfur in amino acids (cysteine and methionine), whereas the inorganic compounds displace sulfur during the synthesis of mercapturic acids and during the reaction of selenites with thiol groups [11,48]. In the result, we can observe distorted, dysfunctional enzymes and protein molecules, which cause the occurrence of disturbances in the biochemical functioning of the cell [49,50]. Symptoms of selenium poisoning cause hair loss and skin and nail lesions [51]. A characteristic symptom of selenium poisoning is the odor of garlic in the exhaled breath because of the presence of a volatile metabolite—dimethyl selenide [1]. Early symptoms of acute poisoning include the occurrence of hypotension and tachycardia. Neurological symptoms include tremor and muscle contractions [52]. Recent studies suggest that increased consumption of selenium may increase the risk of type 2 diabetes [53]. The intake of a large amount of substances, for example, selenic acid, can damage the mucus membrane of the digestive tract, nausea, and diarrhea [54].

Other symptoms of selenium poisoning are anemia, dry cough, fever, and hypersalivation. Poisoning leads to increased permeability of the capillaries and nephrosis [51]. Selenium poisoning is usually chronic and is characterized by an acute course [10]. Toxic symptoms occur in individuals consuming 5 mg selenium per day or among those exposed to air concentration exceeding 0.2 mg/m^3^ [55]. In general, the concentration of selenium in urban air is in the range from 1 to 10 ng/m^3^. An extremely high concentration of >100 ng/m^3^ was observed in Ankara (Turkey) [56].

In the human organism, selenium acts as an antioxidant (e.g., glutathione peroxidase (GPX)), protecting against harmful effect of free radicals. Therefore, covering the demand for this element reduces the risk of cancer [7,13]. Selenium through glutathione reductase and other selenoproteins controls the operation of substances characterized by antioxidant activity, thus playing a key role in the protection of the organism. The share of selenium in the metabolic pathways associated with the protection of cells against oxidative stress causes changes in the activity of selenoproteins. Selenoprotein expression is regulated by the concentration of this element [57]. However, the selenium concentration does not affect the rate of transcription of selenoprotein genes. The observed differences in protein expression are the result of changes in mRNA translation or reduced stability (increased degradation). Twenty-five selenoprotein genes have been identified in sequenced mammalian genomes. Selenium deficiency or excess regulates the transcription of these selenoproteins. Depending on the selenium dose, diverse effects of this element have been observed on cellular functions (immunity, energy metabolism) [58]. Individuals whose blood (serum) selenium level is low with deficiencies of vitamin E accompanied are at the increased risk of developing cancer [59]. In healthy individuals, the level of this element in blood is higher by several percent in comparison with those who suffer from cancer [60].

Selenium affects the functioning of the thyroid gland [61]. Changes in thyroid function resulting from insufficient coverage of demand for this element may result in mood worsening, as well as impairment of behavior and cognitive functions [62]. The effects of these conditions can be alleviated by selenium supplementation [61]. Insufficient supply of this element in the diet decreases the activity of 5′-thyronine deiodinase, which leads to decreased concentration of triiodothyronine in blood. Simultaneous deficiency of iodine and selenium may be the main cause of the underdevelopment of the nervous system caused by congenital hypothyroidism [63]. Selenium increases the activity of macrophages and the production of immunoglobulin. It also increases cytolysis of the so-called NK cells (natural killer) [28,44]. Selenium slows down the aging process by increasing the flexibility of tissues. In addition, it relieves symptoms that occur during menopause [64].

## 6. Importance of Selenium in Biologically Active Compounds

The essential biological importance of selenium is associated with its occurrence in proteins and enzymes. Several selenium-dependent enzymes in which the active center contains selenium in the form of selenocysteine moiety have been identified. The best-characterized selenoenzymes commonly occurring in mammals are glutathione peroxidase, selenoprotein P, and thyroxine 5-deiodinase. Glutathione peroxidase and selenoprotein P catalyze redox reactions [65]. Other enzymatic proteins that are involved in important functions of the organisms are formate dehydrogenase, nicotinic acid hydroxylase, glycine reductase, thiolase, and xanthine dehydrogenase [30].

Glutathione peroxidase (GSH-Px) is the first identified selenoenzyme, consisting of four subunits, each containing a selenium atom in the form of selenocysteine. The latter one is an antioxidant agent, belonging to the so-called free-radical scavengers [66]. This enzyme catalyzes the biosynthesis of glutathione, a tripeptide that plays an important role in protecting organisms against oxidative action of hydrogen peroxide (H_2_O_2_) and organic peroxides [30]. In the presence of glutathione, peroxides are reduced to hydroxyl compounds, that is, alcohol or water. Through the elimination of hydrogen peroxide from the body, this enzyme protects fatty acids, red blood cells, and hemoglobin against oxidation and protects cellular components such as cell membranes and DNA from the destructive effects of oxidation [1,67].

Glutathione peroxidase prevents the so-called oxidative stress that leads to various diseases [30]. Antioxidant properties of both the enzyme and selenium and their protective effect against DNA are used in anti-cancer therapies. Selenium, by neutralizing the negative effect of aflatoxins, reduces their carcinogenic and teratogenic effect and inhibits the growth of cancer cells [68,69]. Potentially, anticarcinogenic mechanisms of selenium interaction relate to the introduction of changes in metabolizing carcinogens, changing the interaction mechanism between carcinogens and DNA, increasing the amount of glutathione, intensification of detoxification processes, selective slowing of the energy metabolism in tumor cells, modification of permeability of cell membranes, and stimulating the immune system of the organism [70,71].

Glutathione reductase is another enzyme that contains selenium. This enzyme catalyzes the process of reducing the oxidized form of glutathione to a reduced form, being involved in the decomposition of organic peroxides and hydrogen [30]. Glutathione reductase is responsible for the maintenance of the appropriate level of reduced glutathione, in order to protect cells from accumulation of peroxide and its damage [66].

Of all the organs in the human body, high selenium content per mass unit can be found in the thyroid gland. Selenium determines proper synthesis, activation, and metabolism of thyroid hormones. It is a component of thyroxine 5-deiodinase. This enzyme is responsible for catalyzation of thyroxine (T_4_) deiodation to its active form known as 3,3,5-triiodothyronine (T_3_) or to inactive form—rT_3_ isomer. Deiodation occurs in peripheral tissues, particularly in kidneys, liver, and skeletal muscles. This process can be deregulated by selenium deficiency in the organism. This indicates the important role of selenium in the proper metabolism of thyroid hormones. In the diagnosis of thyroid diseases, the levels of this element should be considered [61].

Selenium exhibits synergy with vitamin E. In metabolic processes, sulfur amino acids such as cystine and methionine are closely associated with it. The combined interaction of selenium and tocopherol gives the best results in the protection of organs against the destructive effects of free radicals. The combination of these compounds effectively protects mitochondria, cytochrome, and microsomal membranes from the oxidation of fatty acids, determining appropriate growth and fertility. Combined administration of selenium and vitamin E results in an immunostimulatory effect [72].

## 7. Sources of Selenium in the Diet

Selenium is accumulated in the human organism to the largest extent mainly through ingestion. The products of plant and animal origin are the main sources of this element. Plants accumulate selenium in the form of inorganic compounds, selenates (IV) or (VI), which are then converted into organic forms, in particular selenomethionine and selenocysteine. Selenocysteine dominates in the products of animal origin. This is the form in which selenium is consumed by humans, where further conversion of this element occurs, it is bound to amino acids and proteins [73]. The amount of selenium in the diet is diverse and depends on the location in which plants were growing and animals were living [74,75]. It is determined by the amount of selenium in the soil in a given area, soil type, agro-climatic conditions, and the type of crop [53] on the geographical region [74].

The bioavailability of selenium is dependent on many factors, of which the main factor is attributed to the chemical form of this element. Selenium is most easily absorbed in the form of organic compounds and in the presence of vitamins A, D, and E. Bioavailability of selenium contained in foods is also determined by dietary factors such as fat, protein, and heavy metals content [7,31].

Protein rich foods were found to contain higher levels of selenium, whereas low levels were found in plants containing low protein. The main sources of selenium in the diet are foods for example, cereals, meat and dairy products, fishes, seafood, milk, and nuts [31]. A rich source of selenium is found in the sea salt, eggs (only in case of Se-yeast supplementation of feed), giblets, yeast (yeasts containing selenium), bread, mushrooms, garlic, asparagus, kohlrabi (enriched with this element) [1,7,76]. Fruits and vegetables are characterized by a relatively low selenium content. It primarily occurs in the protein fraction, so that plants and vegetables containing a small amount of protein are a poor source of selenium [77]. A large content of selenium can be found in plants (hyperaccumulators), e.g., *Astragalus bisulcatus* and several representatives of *Brassicaceae.* Food products of animal origin are characterized by a diverse amount of selenium, depending on the geographical area in which the animals lived and the supply of selenium in the diet (Table 3).

Marine fishes caught in the north-western area of the Atlantic Ocean contain more selenium (168–825 ng/g) in comparison with freshwater fishes from of west part of the US (143–576 ng/g). Selenium content in chicken eggs is affected to the greatest extent by the diet of hens. Selenium in the egg yolk is present as a phosphoavidin—bound form, whereas in hen’s egg as ovalbumin—bound form. The same products originating from different countries contain different concentrations of selenium [53]. Modified milk for children contains almost four times lesser selenium in comparison with human milk. As a result of feeding infants with modified milk, ~3.5 µg of selenium is daily introduced into the organism, whereas breastfed babies intake ~13.3 µg of this element from the mother’s milk [78].

Cereal products cover ~50% of the daily intake of selenium, whereas the proportion of meat, poultry, and fishes accounts for ~35%. Water and drinks provide 5–25% of selenium. The portion of fruit in meeting the demand for selenium is relatively small and is <10%. Fresh and thermally untreated vegetables provide ~11% of selenium in a properly balanced diet. Thermal processing of food products can lead to loss of selenium in the food because of the formation of volatile selenium compounds. These losses are significant and can reach tens of percent. Bioavailability of selenium in food is dependent on the form of its occurrence and the content of such compounds as protein, fat, and heavy metals. The bioavailability of selenium is reduced in the presence of heavy metals and sulfur but increases in the presence of vitamins A, C, E, and low-molecular-weight proteins containing methionine [1,81].

## 8. Selenium Supplementation

The primary source of selenium is the appropriately selected and balanced diet, covering the demand for this element. Selenium deficiency in healthy individuals results from a low content of this element in food or consumption of products with poor selenium content. It should be emphasized that the proportion of selenium present in a daily dose of individual food products is diverse [1]. Due to the widespread deficiency of selenium in humans, the need for supplementation of this element calls attention. Introduction of this element into the human body can occur indirectly, through the addition of selenium to fertilizers or fodder used to feed animals [85,86]. Among direct supplementation methods, we can include the use of food supplements containing vitamins and micronutrients [31].

A country that introduced selenium supplementation due to a low concentration of selenium in the environment and too low supplementation of this element in the diet of people was Finland [85,86,87]. Effects of selenium supplementation can be observed after a few weeks. They are dependent on the degree of deficiency of selenium in the organism, its dose, and chemical form. The reaction of the organism is more quickly noticed for supplementation of the organic form: selenomethionine in comparison with inorganic forms of selenium [88]. The most direct supplementation method is an individual use of properly enriched food supplements usually in the form of preparations, with in inorganic source of selenium or supplements based on selenium yeast biomass [89]. Higher bioavailability and greater safety of preparations containing organic selenium differ from those with a content of inorganic selenium salts [88]. Additional advantages are low cost and simple manufacturing process of yeast biomass rich in selenium. Accordance with EU regulations the production of dietary supplements containing selenium may utilize dietary supplements in the form of selenium-enriched yeasts (<2500 μg S/g) [90].

Selenomethionine, which is a dominant form in selenium yeast formulations, consists of a source of selenium in proteins; therefore, its use is a preferred strategy in preventing from deficiencies of this element in humans and animals. In contrast, preparations containing inorganic forms as sodium selenite may be more beneficial in clinical procedures (in cancer), in which fast effect is a priority [43]. The use of supplements containing organic selenium of yeast origin, in case of deficiency, exhibits a multidirectional beneficial effect on human health [91]. An interesting supplementation strategy is to use functional food products. These can be products obtained from selenium-enriched plant biomass. Fermented foods contain lactic acid bacteria accumulating significant quantities of selenium, similar to yeast. Selenium supplementation in diet is recommended in the treatment of pancreatitis, infertility, and asthma [69,92]. Reduced supply of selenium occurs also during other diet-related diseases. This relates to parenterally fed patients with malabsorption and those suffering from metabolic diseases. Patients undergoing a specialized chemical treatment and after radiotherapy are also at risk of selenium deficiency [1]. Selenium deficiencies may be treated by the use of appropriately balanced pharmaceutical preparations (medicines) and dietary supplements that include selenium. These preparations are available without prescription. Selenium contained in these formulations may be present in organic form, for example, selenomethionine, or in the inorganic form of selenites (IV) and selenates (VI). Commonly available preparations are characterized by multidirectional action. Apart from supplementing the deficiency of selenium, they have a beneficial effect on the cardiovascular system and thyroid function. It positively affects the process of treatment of cardiovascular diseases and potentially reduces the risk of developing certain types of cancer [68].

Selenium in the forms of both inorganic compounds (Na_2_SeO_3_ and Na_2_SeO_4_) and organic ones (selenocysteine, selenomethionine, and selenoglutathione) introduced into the organism through food or drinking water results in a significant reduction of chemically induced cancers [93]. Protective effect of selenium occurs after crossing a threshold amount in a diet, which corresponds to a dose of 250–300 µg Se/day. From health-improving viewpoint, it is important that a protective effect occurs in organs that are the main locations of tumors in humans (stomach, intestine, mammary gland, and liver) [69].

It has been demonstrated that supplementation of fodder with selenium effectively protects against the increase of blood pressure in animals regularly exposed to heavy metal poisoning. Selenium supplementation reduces necrotic lesions in the testis and in animal fetus [94]. In animals exposed to adverse effects of mercury compounds increased amounts of selenium reduce the formation of necrotic lesions in kidneys. Selenium plays a role of chelating agent of heavy metals by the formation of toxic selenium–metal complexes [3]. The most valuable and safe supplementation is the one that uses preparations containing selenium yeast. Their use has a multidirectional and beneficial effects on human health [95]. Organic selenium derived from yeast has a better bioavailability in comparison with their inorganic forms [96]. The use of selenium yeast on a large scale can help to reduce the deficiencies of this element resulting from a diet with low selenium content [97].

Fear against the introduction of selenomethionine (SeMet) into proteins, which could lead to the achievement of toxic levels of selenium, is not justified because of the natural equilibrium established, which prevents the uncontrolled accumulation of selenium in the organism. Furthermore, the release of SeMet from proteins through metabolic processes that occur during the disease should not result in toxic effect of selenium, because, until now, the mechanism that could be responsible for the selective release of SeMet during catabolism has not been identified [88].

Attention is paid to alternative forms of supplementation. In addition to yeast, biomass of bacteria and plants can also be enriched in selenium. Fermenting lactic acid bacteria supports excessive accumulation of selenium; thus, the concept of using microorganisms for the production of functional food is justified. Sourdough bakery with starter cultures enriched in selenium can constitute an example [1]. Research is being conducted on obtaining plants enriched in selenium and thus the possibility of obtaining protein fractions and selenium-enriched food products from them [1,77]. Many plants are a poor source of selenium; however, they exhibit the ability to accumulate this element in the cultivation and convert it to appropriate forms, thus becoming its potential reservoir [73].

## 9. Conclusions

Consumer lifestyle changes are a stimulus to creating new nutritional solution. Selenium plays an important role in maintaining the homeostasis of the human body. The issue of selenium supplementation in order to prevent various disorders is still an open question and requires further research. Due to promising data from clinical trials, the use of supplementation of this element should be considered. An innovative product, an example of which may be dietary supplements enriched in selenium should be more adapted to the requirements of consumers. Yeast enriched with organic selenium forms can be a good source of this ingredient to the consumer’s diet. However, it should be remembered that every product should be precisely checked for the content of individual nutrients, safety, and correctness of action. Satisfying the sophisticated consumer needs with simultaneous supplementation of deficient elements can be an effective tool in the fight for health.

## Figures and Tables

**Table 1 molecules-24-01298-t001:** Selected physical and chemical properties of selenium.

Properties	Selenium
Electronic configuration	[Ar] 3d^10^4s^2^4p^4^
Atomic number	34
Atomic weight	78.96
Density [g/cm^3^]	4.808
Melting temperature (°C)	220
Boiling temperature (°C)	685
Oxidation states	−II, 0, IV, VI
Electron affinity	−4.2 Ev
Ionization potential	9.75 Ev

**Table 2 molecules-24-01298-t002:** Recommended dietary allowances (RDAs) for selenium [26,32,33].

Age (years)	Males and Females, Selenium (μg/day)
1–3	15–20
4–13	30–40
14–50	55–70
51+	70–100

**Table 3 molecules-24-01298-t003:** Selenium content in selected food products.

Food	Selenium Content (μg/g)	References
Brazil nuts	0.2–512	[78]
Yeast	500–4000	[1,7,74]
Bread	0.09–0.20	[77,79]
Fish	0.06–0.63	[74,79,80,81]
Eggs	0.09–0.25	[74,82,83]
Chicken	0.15	[79]
Beef	0.01–0.73	[74,83]
Pork	0.27–0.35	[77,79]
Broccoli	0.012	[83]
Milk	0.01–0.06	[79,80]
Chocolate	0.04	[74,79,80,81,83]
Liver	0.3–0.4	[84]
Beef kidney	1.45	[81]

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
