# Peer review of "Selenium–Fascinating Microelement, Properties and Sources in Food"

_molecules, 2019, doi:10.3390/molecules24071298_

Round 1

Reviewer 1 Report

Comments and Suggestions for Authors:

This manuscript sets out to review recent progress in understanding the properties and effects of selenium as well as its sources in feed/food, recommending ways of supplementation for optimal health. This is an area of considerable interest, but has also been the subject of a number of other recent review articles (see, for example, the special issues of Antioxidants and Redox Signaling on this topic). There is therefore a lot of previous, and recent, information currently in the literature. This said there are a number of recent papers which have published in the last three years. However, most of the refereed articles used (80%) were published before 2016.

However, there are places in the text where recent papers are discussed but there is little critical appraisal or synthesis of the data in to a more comprehensive picture.

The paper is dense, sometimes unconnected, and rather repetitive in places (e.g. there are multiple sections that deal with inorganic vs. organic Se) and the manuscript could easily be given greater clarity and focus by revising the data presented.

The parts on the reactions with oxidants is rather disappointing – the discussion is brief and mainly focused on the reactions of radicals with lipids and DNA – there is little discussion of reactions with proteins (which are the most abundant targets in most biological systems) and are of particular relevance given that many of the selenoproteins are not membrane bound.

The manuscript should be read and revised by a native English speaker as there are a number of phrases and sentences that are difficult to understand at present.

Minor points:

Line 49: Please delete “selenium”

Line 71: Please correct “are stable”

Line 77: Please rewrite “It selenium is emitted…”

Line 97: (<0.1 mg/kg)?

Line 148: Please add “present in that region.”

Line 180: Please correct “which cause the occurrence”

Line 188: Please put period [.]

Lines 204-205: Please correct “as well as impairment of behavior and…”

Lines 219-222: Why are these specific proteins chosen here?

Line 237: Please put comma [,] “metabolizing carcinogens, changing interaction”

Line 257: Please correct “protects mitochondria”

Line 261: Please correct numbering of sections from this point on

Line 301: Please add “selenium in food”

Line 344: Please correct “cardiovascular system and thyroid function”

Line 349: Does Ref [85] correspond to this citation?

Lines 365-370: Please rewrite and give more Literature References. Moreover, an elusive mechanism does not guarantee the absent of an effect.

Line 382: Please delete “the world will allow to explore”

Author Response

Thank you for considering publishing our article entitled „Selenium – fascinating microelement, properties and sources in food” in Molecules and for referring the manuscript to revision (Manuscript ID molecules-472014)

      We thank Reviewers' for taking the time to review the article. We appreciate all constructive comments, questions and suggestions which improved the work and are valuable indications for the future also.

    Reviewers' suggestions were taken into account and changes are highlighted within the manuscript using colored text (yellow for Reviewers).

     Referring Reviewer’s comments and questions the answers and corrections are presented below.

      We hope that all responses to the Reviewer’s remarks would be comprehensive and sufficient and that the revised manuscript could be accepted for publication in Molecules.

We are looking forward to receiving your final decision.

Yours sincerely,

Marek Kieliszek

Reviewer 1

The manuscript was corrected according to Reviewer’s indications and remarks concerning:

This manuscript sets out to review recent progress in understanding the properties and effects of selenium as well as its sources in feed/food, recommending ways of supplementation for optimal health. This is an area of considerable interest, but has also been the subject of a number of other recent review articles (see, for example, the special issues of Antioxidants and Redox Signaling on this topic). There is therefore a lot of previous, and recent, information currently in the literature. This said there are a number of recent papers which have published in the last three years. However, most of the refereed articles used (80%) were published before 2016.

Many scientists in the world deal with the topic of selenium. It is very important to expand the available knowledge with new information. Of course, there are new articles about the use, the characteristics of selenium in many aspects of life. I agree with the reviewer.

The list of references was corrected according to Reviewer remarks:

Zhang, X. W My Element: Selenium. Chem. Eur. J. 2019, 25(11), 2649-2650. doi:10.1002/chem.201804075

Duntas, L.H., Benvenga, S. Selenium: an element for life. Endocr. 2015, 48(3), 756-775. doi:10.1007/s12020-014-0477-6

Reich, H.J.; Hondal, R.J. Why nature chose selenium? ACS Chem. Biol. 2016, 11(4), 821-841. doi:10.1021/acschembio.6b00031

Mason, R.P.; Soerensen, A.L.; DiMento, B.P.; Balcom, P.H. The global marine selenium cycle: insights from measurements and modeling. Global Biogeochem. Cycles 2018, 32(12), 1720-1737. doi: 10.1029/2018GB006029

Shahid, M.; Niazi; N.K.; Khalid, S.; Murtaza, B.; Bibi, I.; Rashid, M.I. (2018). A critical review of selenium biogeochemical behavior in soil-plant system with an inference to human health. Environ. Pollut. 2018, 234, 915-934. doi:10.1016/j.envpol.2017.12.019

He, Y.; Xiang, Y.; Zhou, Y.; Yang, Y.; Zhang, J.; Huang, H.; Shang, C.; Luo, L.; Gao, J.; Tang, L. Selenium contamination, consequences and remediation techniques in water and soils: A review. Environ. Res. 2018, 164, 288-301. doi:10.1016/j.envres.2018.02.037

WHO. Selenium in drinking-water. Background document for preparation of WHO Guidelines for drinking-water quality. Geneva: World Health Organization (WHO/SDE/ WSH/03.04/13), 2003.

Post, M.; Lubiński, W.; Lubiński, J.; Krzystolik, K.; Baszuk, P.; Muszyńska, M.; Marciniak, W. (2018). Serum selenium levels are associated with age-related cataract. Ann. Agric. Environ. Med. 2018, 25(3), 443-448. doi: 10.26444/aaem/90886

Kipp, A.P.; Strohm, D.; Brigelius-Flohé, R.; Schomburg, L.; Bechthold, A.; Leschik-Bonnet, E.; Heseker, H.; DGE, G.N.S. Revised reference values for selenium intake. J. Trace Elem. Med. Bio. 2015, 32, 195-199. doi:10.1016/j.jtemb.2015.07.005

Hsueh, Y.M.; Su, C.T.; Shiue, H.S.; Chen, W.J.; Pu, Y.S.; Lin, Y.C.; Tsai, C.S.; Huang, C.Y. Levels of plasma selenium and urinary total arsenic interact to affect the risk for prostate cancer. Food Chem. Toxicol. 2017, 107, 167-175. doi: 10.1016/j.fct.2017.06.031

EFSA Panel on Additives and Products or Substances used in Animal Feed (FEEDAP), Bampidis, V.; Azimonti, G.; Bastos, M.D.L.; Christensen, H.; Dusemund, B.; Kouba, M.; Durjava, M.K.; López‐Alonso, M.; López Puente, S.; Marcon, F.; Mayo, B.; Pechová, A.; Petkova, M.; Ramos, F.; Sanz, Y.; Villa, R.; Woutersen. R.; Cubadda, F.; Flachowsky, G.; Gropp, J.; Leng, L.; López‐Gálvez, G.; Marcon, F. Assessment of the application for renewal of authorisation of selenomethionine produced by Saccharomyces cerevisiae CNCM I3060 (selenised yeast inactivated) for all animal species. EFSA J. 2018, 16(7), e05386. doi:10.2903/j.efsa.2018.5386

Institute of Medicine, Food and Nutrition Board. Dietary Reference Intakes for vitamin C, vitamin E, selenium, and carotenoids. Washington, DC: National Academy Press; 2000, 1–20.

Strand, T.A.; Lillegaard, I.T.L.; Frøyland, L.; Haugen, M.; Henjum, S.; Løvik, M.; Stea T.H.; Holvik, K. Assessment of selenium intake in relation to tolerable upper intake levels. Eur. J. Nut. Food Saf. 2018, 155-156. doi:10.9734/EJNFS/2018/42536

Manzanares, W.; Hardy, G. Can dietary selenium intake increase the risk of toxicity in healthy children? Nutrition 2016, 32(1), 149-150. doi:10.1016/j.nut.2015.07.001

WHO/FAO/IAEA. Trace elements in human nutrition and health. Geneva, Switzerland: World Health Organization; 1996

Murdolo, G.; Bartolini, D.; Tortoioli, C.; Piroddi, M.; Torquato, P.; Galli, F. Selenium and cancer stem cells. Adv. Cancer Res. 2017, 136, 235–257. doi: 10.1016/bs.acr.2017.07.006

Okunade, K.S.; Olowoselu, O.F.; Osanyin, G.E.; JohnOlabode, S.; Akanmu, S.A.; Anorlu, R.I. Selenium deficiency and pregnancy outcome in pregnant women with HIV in Lagos, Nigeria. Int. J. Gynaecol. Obstet. 2018, 142(2), 207-213.doi:10.1002/ijgo.12508

Puccinelli, M.; Malorgio, F.; Pezzarossa, B. Selenium enrichment of horticultural crops. Molecules 2017, 22(6), 933. doi:10.3390/molecules22060933

Song, M.; Kumaran, M.N.; Gounder, M.; Gibbon, D.G.; Nieves-Neira, W.; Vaidya, A.; Hellmann, M.; Kane, M.P.; Buckley, B.; Shih, W.; et al. Phase I trial of selenium plus chemotherapy in gynecologic cancers. Gynecol. Oncol. 2018. 150(3), 478-486. doi: 10.1016/j.ygyno.2018.07.001

Tortelly, V.C.; Melo, D.F.; Matsunaga, A.M. The relevance of selenium to alopecias. Int. J. Trichology 2018, 10(2), 92-93. doi: 10.4103/ijt.ijt_37_17

Lippman, S.M.; Klein, E.A.; Goodman, P.J.; Lucia, M.S.; Thompson, I.M.; Ford, L.G.; Parnes H.L.; Minasian, L.M.; Gaziano, J.M.; Hartline, J.A.; et al. Effect of selenium and vitamin E on risk of prostate cancer and other cancers: the Selenium and Vitamin E Cancer Prevention Trial (SELECT). Jama 2009, 301(1), 39-51. doi:10.1001/jama.2008.864

Ventura, M.; Melo, M.; Carrilho, F. Selenium and thyroid disease: From pathophysiology to treatment. Int. J. Endocrinol., 2017, Article ID 1297658. doi: 10.1155/2017/1297658

Mondal, S.; Mugesh, G. Novel thyroid hormone analogues, enzyme inhibitors and mimetics, and their action. Mol. Cell. Endocrinol. 2017, 458, 91-104. doi:10.1016/j.mce.2017.04.006

Brigelius-Flohé, R.; Flohé, L. Selenium and redox signaling. Arch. Biochem. Biophys. 2017, 617, 48-59. doi:10.1016/j.abb.2016.08.003

Cai, Z.; Zhang, J.; Li, H. Selenium, aging and aging-related diseases. Aging. Clin. Exp. Res. 2018, 1-13. doi:10.1007/s40520-018-1086-7

Hatfield, D.L.; Carlson, B.A.; Tsuji, P.A.; Tobe, R.; Gladyshev, V.N. Selenium and cancer. In Molecular, Genetic, and Nutritional Aspects of Major and Trace Minerals. Academic Press 2017, 463-473. doi:10.1016/B978-0-12-802168-2.00038-5

Woo, J.; Lim, W. Anticancer effect of selenium. Ewha Med. J. 2017, 40(1), 17-21. doi: 10.12771/emj.2017.40.1.17

Lammi, M.; Qu, C. Selenium-Related transcriptional regulation of gene expression. Int. J. Mol. Sci. 2018, 19(9), 2665. doi:10.3390/ijms19092665

dos Santos, M.; da Silva Júnior, F.M.R.; Muccillo-Baisch, A.L. Selenium content of Brazilian foods: a review of the literature values. J. Food Compost. Anal. 2017, 58, 10-15. doi:10.1016/j.jfca.2017.01.001

Pavlovic, Z.; Miletic, I.; Zekovic, M.; Nikolic, M.; Glibetic, M.. Impact of selenium addition to animal feeds on human selenium status in Serbia. Nutrients 2018, 10(2), 225. doi:10.3390/nu10020225

Lönnerdal, B.; Vargas-Fernández, E.; Whitacre, M. Selenium fortification of infant formulas: does selenium form matter? Food Funct. 2017, 8(11), 3856-3868. doi:10.1039/C7FO00746A

Regulation (EC) 1170/2009 of 30 November 2009 amending Directive 2002/46/EC of the European Parliament and of Council and Regulation (EC) No 1925/2006 of the European Parliament and of the Council as regards the lists of vitamin and minerals and their forms that can be added to foods, including food supplements. Off. J. Eur. Union 2009, 38, 232–238.

Brigelius-Flohé, R. Selenium in Human Health and Disease: An Overview. In: Michalke B. (eds) Selenium. Molecular and Integrative Toxicology. Springer, Cham, 2018, 3-26. doi:10.1007/978-3-319-95390-8_1

EFSA. Selenium-enriched yeast as source for selenium added for nutritional purposes in foods for particular nutritional uses and foods (including food supplements) for the general population-scientific opinion of the panel on food additives, flavourings, processing aids and materials in contact with food. EFSA J., 2008, 766, 1–42

However, there are places in the text where recent papers are discussed but there is little critical appraisal or synthesis of the data in to a more comprehensive picture.

All comments are presented in the article. New information has been added. Article is a collection of all the important information about selenium. Of course the article would contain more information but then it would be too long within the journal.

The paper is dense, sometimes unconnected, and rather repetitive in places (e.g. there are multiple sections that deal with inorganic vs. organic Se) and the manuscript could easily be given greater clarity and focus by revising the data presented.

I do not agree with the reviewer. The article is divided into chapters (Physicochemical properties of selenium, Occurrence of selenium in the environment, Selenium in human organism, Effect of deficiency and excess of selenium on human health, Importance of selenium in biologically active compounds, Sources of selenium in the diet, Selenium supplementation). Perhaps the information is very much in the article that's why such an opinion.

The presented data is balanced with information on organic and inorganic selenium in individual chapters. Changes in the text have a yellow color.

The parts on the reactions with oxidants is rather disappointing – the discussion is brief and mainly focused on the reactions of radicals with lipids and DNA – there is little discussion of reactions with proteins (which are the most abundant targets in most biological systems) and are of particular relevance given that many of the selenoproteins are not membrane bound.

I've added new information:

The share of selenium in the metabolic pathways associated with the protection of cells against oxidative stress causes changes in the activity of selenoproteins. Selenoprotein expression is regulated by the concentration of this element [55]. However, selenium concentration does not affect the rate of transcription of selenoprotein genes. The observed differences in protein expression are the result of changes in mRNA translation or reduced stability (increased degradation). Twenty-five selenoprotein genes have been identified in sequenced mammalian genomes. Selenium deficiency or excess regulates the transcription of these selenoproteins. Depending on selenium dose, diverse effects of this element have been observed on cellular functions (immunity, energy metabolism) [56].

Selenium plays an important role in maintaining the homeostasis of the human body. The issue of selenium supplementation in order to prevent various disorders is still an open question and requires further research. Due to promising data from clinical trials, the use of supplementation of this element should be considered.

The manuscript should be read and revised by a native English speaker as there are a number of phrases and sentences that are difficult to understand at present.

All minor notes have been included. The manuscript has been corrected by a native speaker (www.translmed.com).

Reviewer 2 Report

I find the article useful. It is a useful review and deserves to be published. My main points are: --- the text needs some improvement (see the highlights in the attached PDF) --- there is some repetition (e.g., on the presence of Se in the environment and in food, its biochemical and toxicological role ...), which results from the structure adopted. But this is not a serious problem. --- there is some confusion with the regulatory classification of products containing Se that are available in the market (food supplements, medicines, other ...). The author should review the EU legislation on Food Supplements. Namely to see the chemical forms that are authorized for incorporation as ingredients; --- My major point is related to bibliographical references. I give only a few examples: 1) To support the statement that "The World Health Organization (WHO) recommends a daily dose of selenium at a level of 55 μg for adults," the authors use an article entitled "Characterization of selenium speciation in selenium-enriched button mushrooms (Agaricus bisporus) and selenized yeasts (dietary supplement) using X-ray absorption near-edge structure (XANES) spectroscopy. This is not acceptable, even more so in a review article. The author must use the original reference --- i.e., the WHO document. 2) To support "The use of selenium yeast on a large scale can help reduce the deficiencies of this element resulting from a diet with low selenium", the author uses an article about the "Effects of selenium on morphological changes in Candida utilis ATCC 9950 yeast cells". This is not good policy. 3) To support that "Selenium (...) introduced into the organism through food or drinking water results in a significant reduction of chemically induced cancers" (a very, very "strong" statement!), the author use an article entitled entitled "Effect of selenium on lipid and amino acid metabolism in yeast cells". This is not acceptable! ... Finally the author refers to the SELECT trial to support that "Individuals whose blood selenium level is low with vitamin deficiencies and accompanied at the increased risk of developing cancer", but does not refer to its major conclusion: "no reduction in risk of prostate (or vitamin E) supplementation". In the case of Se, at 200 μg / d as L-selenomethionine. In my opinion, Se is a very important element, no doubts. And there is good evidence of the negative effects of Se deficiency. But, on the contrary, there is no solid evidence of the positive effects of "supplementation" above the recommended / normal daily intake.

Author Response

Thank you for considering publishing our article entitled „Selenium – fascinating microelement, properties and sources in food” in Molecules and for referring the manuscript to revision (Manuscript ID molecules-472014).

   We thank Reviewers' for taking the time to review the article. We appreciate all constructive comments, questions and suggestions which improved the work and are valuable indications for the future also.

  Reviewers' suggestions were taken into account and changes are highlighted within the manuscript using colored text (yellow for Reviewers).

  Referring Reviewer’s comments and questions the answers and corrections are presented below.

  We hope that all responses to the Reviewer’s remarks would be comprehensive and sufficient and that the revised manuscript could be accepted for publication in Molecules.

We are looking forward to receiving your final decision.

Yours sincerely,

Marek Kieliszek

The manuscript was corrected according to Reviewer’s indications and remarks concerning:

I find the article useful. It is a useful review and deserves to be published. My main points are: the text needs some improvement (see the highlights in the attached PDF)

Thank you for your opinion. All comments are included in the article

there is some repetition (e.g., on the presence of Se in the environment and in food, its biochemical and toxicological role ...), which results from the structure adopted. But this is not a serious problem.

The change results from the presence of selenium in various food products found in poor regions (low concentration) in this element. The deficiencies of selenium are a critical problem worldwide, with the negative impact on health and lifespan. Biofortification is the process by which the nutritional quality of food is improved through agricultural approach that can improve human nutrition on the world. Biofortification of agricultural crops with Se, by means of adding Se along with fertilizers, is a useful technique to increase the consumption of Se by animals and man.

there is some confusion with the regulatory classification of products containing Se that are available in the market (food supplements, medicines, other ...). The author should review the EU legislation on Food Supplements. Namely to see the chemical forms that are authorized for incorporation as ingredients;

New information has been added, supplemented with data (eg review of EU legislation on Food Supplements):

Regulation (EC) 1170/2009 of 30 November 2009 amending Directive 2002/46/EC of the European Parliament and of Council and Regulation (EC) No 1925/2006 of the European Parliament and of the Council as regards the lists of vitamin and minerals and their forms that can be added to foods, including food supplements. Off. J. Eur. Union 2009, 38, 232–238.

EFSA. Selenium-enriched yeast as source for selenium added for nutritional purposes in foods for particular nutritional uses and foods (including food supplements) for the general population-scientific opinion of the panel on food additives, flavourings, processing aids and materials in contact with food. EFSA J., 2008, 766, 1–42

EFSA Panel on Additives and Products or Substances used in Animal Feed (FEEDAP), Bampidis, V.; Azimonti, G.; Bastos, M.D.L.; Christensen, H.; Dusemund, B.; Kouba, M.; Durjava, M.K.; López‐Alonso, M.; López Puente, S.; Marcon, F.; Mayo, B.; Pechová, A.; Petkova, M.; Ramos, F.; Sanz, Y.; Villa, R.; Woutersen. R.; Cubadda, F.; Flachowsky, G.; Gropp, J.; Leng, L.; López‐Gálvez, G.; Marcon, F. Assessment of the application for renewal of authorisation of selenomethionine produced by Saccharomyces cerevisiae CNCM I3060 (selenised yeast inactivated) for all animal species. EFSA J. 2018, 16(7), e05386. doi:10.2903/j.efsa.2018.5386

Institute of Medicine, Food and Nutrition Board. Dietary Reference Intakes for vitamin C, vitamin E, selenium, and carotenoids. Washington, DC: National Academy Press; 2000, 1–20.

WHO/FAO/IAEA. Trace elements in human nutrition and health. Geneva, Switzerland: World Health Organization; 1996

I add new information about selenium:

The share of selenium in the metabolic pathways associated with the protection of cells against oxidative stress causes changes in the activity of selenoproteins. Selenoprotein expression is regulated by the concentration of this element [55]. However, selenium concentration does not affect the rate of transcription of selenoprotein genes. The observed differences in protein expression are the result of changes in mRNA translation or reduced stability (increased degradation). Twenty-five selenoprotein genes have been identified in sequenced mammalian genomes. Selenium deficiency or excess regulates the transcription of these selenoproteins. Depending on selenium dose, diverse effects of this element have been observed on cellular functions (immunity, energy metabolism) [56].

Selenium plays an important role in maintaining the homeostasis of the human body. The issue of selenium supplementation in order to prevent various disorders is still an open question and requires further research. Due to promising data from clinical trials, the use of supplementation of this element should be considered.

The manuscript should be read and revised by a native English speaker as there are a number of phrases and sentences that are difficult to understand at present.

All minor notes have been included. The manuscript has been corrected by a native speaker.

Nowe informacje

My major point is related to bibliographical references. I give only a few examples: 1) To support the statement that "The World Health Organization (WHO) recommends a daily dose of selenium at a level of 55 μg for adults," the authors use an article entitled "Characterization of selenium speciation in selenium-enriched button mushrooms (Agaricus bisporus) and selenized yeasts (dietary supplement) using X-ray absorption near-edge structure (XANES) spectroscopy. This is not acceptable, even more so in a review article. The author must use the original reference --- i.e., the WHO document. 2) To support "The use of selenium yeast on a large scale can help reduce the deficiencies of this element resulting from a diet with low selenium", the author uses an article about the "Effects of selenium on morphological changes in Candida utilis ATCC 9950 yeast cells". This is not good policy. 3) To support that "Selenium (...) introduced into the organism through food or drinking water results in a significant reduction of chemically induced cancers" (a very, very "strong" statement!), the author use an article entitled entitled "Effect of selenium on lipid and amino acid metabolism in yeast cells". This is not acceptable! ... Finally the author refers to the SELECT trial to support that "Individuals whose blood selenium level is low with vitamin deficiencies and accompanied at the increased risk of developing cancer", but does not refer to its major conclusion: "no reduction in risk of prostate (or vitamin E) supplementation". In the case of Se, at 200 μg / d as L-selenomethionine. In my opinion, Se is a very important element, no doubts. And there is good evidence of the negative effects of Se deficiency. But, on the contrary, there is no solid evidence of the positive effects of "supplementation" above the recommended / normal daily intake.

All comments were added to the manuscript.

Scientific work done by Prof. Boguslaw Lipinski of Harvard Medical School showed that selenium can be very important in the fight against cancer. Research works are constantly being continued on this topic. The research topic needs to be developed all the time. An example is a new selenium compound (Selol): Using it in treatment may decrease the risk of lifestyle diseases such as: cancer, cardiovascular diseases, multiple sclerosis, diabetes, rheumatism and many other diseases caused by oxidative stress and redox state disorder in cells. Studies presented by Flis et al. showed that selol is non-toxic and non-mutagenic. It exhibits strong anti-cancer activity in vitro in many cancer cell lines. This preparation of hope for patients for whom standard chemotherapy regimens are ineffective. The potential to use Selol as a prophylactic and therapeutic source of selenium requires further work. The preparation is currently thoroughly studied in several research centers in Poland.

Of course, there is not slimy evidence because clinical trials are underway for some forms of this element (selenium). The test results have not been patented.

The relationship between selenium and the occurrence of cancer has been verified in studies:

Przybylik-Mazurek, E., Zagrodzki, P., Kuźniarz-Rymarz, S., & Hubalewska-Dydejczyk, A. (2011). Thyroid disorders—assessments of trace elements, clinical, and laboratory parameters. Biological trace element research141(1-3), 65-75.

Shen, F., Cai, W. S., Li, J. L., Feng, Z., Cao, J., & Xu, B. (2015). The association between serum levels of selenium, copper, and magnesium with thyroid cancer: a meta-analysis. Biological trace element research167(2), 225-235.

O’Grady, T. J., Kitahara, C. M., DiRienzo, A. G., & Gates, M. A. (2014). The association between selenium and other micronutrients and thyroid cancer incidence in the NIH-AARP Diet and Health Study. PLoS One9(10), e110886.

It should also be noted that low selenium concentration may not be the cause but the effect of general systemic diseases, including cancer.

Reviewer 3 Report

The review does not include all necessary information and should be improved

Author Response

The manuscript was corrected according to Reviewer’s suggestions concerning:

The authors pay much attention to Se supplements- but the title of the review deals only with food products

We believe that the title corresponds to the content of the article. Of course, we agree with the reviewer. But all information about this element can not be attached. The article would be too broad

2) Not complete information is given about chemical forms of Se in food. In particular- Se-containing sugars and small molecular weight (S-analogs) and especially methylated forms of amino-acids typical for Allium cepa and Brassica plants with pronounced anti-carcinogenic activity

Line 273-282, This is a brief description of selenium compounds found in animal and plant products. The wider characterization of selenium compounds is presented in another article: Kieliszek, M.; Błażejak, S. Current knowledge on the importance of selenium in food for living organisms: a review. Molecules 2016, 21, 609.

Selenium hypericumulators are indicator plants of selenium soils and can accumulate in the tissues of organs from a thousand to several thousand milligrams of Se. Selenium constituting over 0.6% of dry biomass of Astragalus biscucatus. Examples of plants of this type are some species: Aster, Atriplex, Comandra, Melilotus, Brassica.

3) As far as food products are concerned the author should mention Se-enriched sprouts as an important functional food and give wider information of Se-eggs (that is feed supplementation with Se-yeast)

The information has been added to the article

4) It is necessary to empathize that changing the import of cereals from endemic regions of the world may significantly decrease the human selenium status (Great Britain)

Of course, the import of cereals can affect the change of the selenium status in the bodies of people and animals.

Selenium deficiencies in humans and animals can be supplemented with various supplements which in recent years have been advertised and recommended. But it would be more appropriate and less expensive to supplement this element in the diet.

The information fully justifies research into the accumulation of selenium by plants

5) Nothing is said that animal organs (liver and kidney) are especially rich with Se

The information has been added to the article

6) Much is spoken about the toxicity of Se but nothing of the possibility of toxicity as a result of mistakes in Se-doses during production of Se-supplements (USA data). In this respects vegetables supplemented with Se seems to be safer compared to artificial supplements as plants may be considered as a buffer between soil Se and humans

Of course they can be safe but so far it has not been possible to identify all forms of the occurrence of this element in plant products (Orbitrap LC-MS, ICP-MS, UPLC-MS analysis). More attention should be paid to making accurate analyzes. Yeast biomass can be obtained faster after 24 hours from 1 liter, we get 13 grams of biomass. Every vegetable and yeast biomass obtained must be checked reliably.

Se-rich yeast has been established as the basic ingredient of Se food and feed supplements. The optimization of the fermentation process towards the full conversion into the desired species (usually selenomethionine), the need of the product characterization in terms of selenium species present and the questions about the traceability of the industrial products are still on the agenda and are a driving force of the developments of suitable analytical methodologies

7) Line 106- speaking about the essentiality of Se it is necessary to indicate that a) SeCys is encoded genetically in mammals and b) Se- is not an essential element for plants but is a powerful antioxidant in all organisms including plants, c) biofortification of plants with Se results in the increase of antioxidants content

We believe that selenium is needed for the functioning of organisms. Of course, the content of this element affects the increase in the level of antioxidant enzymes that defend against ROS. The data has been described in chapter 6.

8) Line 278- “A rich source of selenium is found in the sea salt, salt from the salt mine, eggs, yeast, bread, mushrooms, tomatoes, garlic, asparagus, kohlrabi, and nuts” - ‘salt from the salt mine”- often contains only trace amounts of Se; -“, garlic, asparagus, kohlrabi”- grown on soil with low Se bioavailability or low soil Se contain low Se concentrations -“eggs”- only in case of Se-yeast supplementation of feed -“yeast”- only Se-enriched yeast -“bread”- only from grain rich in Se -“mushrooms”- not all of them, exclusively Boletus, Agaricus, etc -in ordinary conditions of vegetation “tomatoes” contain only trace amount of Se In this respect nothing is said about plants- hyper accumulators of Se (Astragalus for example)

The information has been added to the article

Round 2

Reviewer 1 Report

The revised version of the manuscript is substantially improved over the previous one and most of the points raised have been addressed, so it is now acceptable for publication.

Author Response

I thank Reviewer for taking the time to review the article. I appreciate all constructive comments, questions and suggestions which improved the work and are valuable indications for the future also.

Reviewer 3 Report

There are still some drawbacks in the review. See attached file

Author Response

I thank Reviewer for taking the time to review the article. I appreciate all constructive comments, questions and suggestions which improved the work and are valuable indications for the future also.

Reviewer suggestions were taken into account and changes are highlighted within the manuscript using colored text (yellow for Reviewers). 

Referring Reviewer comments and questions the answers and corrections are presented below.

1)  Line 193 “In general, the concentration of selenium in urban air is in the range from 1 to 10 mg/m3. Extremely high concentration of >100 mg/m3 was observed in Ankara (Turkey) [54]” Reference [54]-is a Chinese work, but not a Turkish one where Se content in air is 43-58 ng/m3. Think that the above sentence deals with ng/m3 but not mg/m3

I added new citation:

EFSA Panel on additives and products or substances used in animal feed (EFSA FEEDAP Panel), Bampidis, V., Azimonti, G., Bastos, M. D. L., Christensen, H., Dusemund, B.; et al. Assessment of the application for renewal of authorisation of selenomethionine produced by Saccharomyces cerevisiae NCYC R397 for all animal species. EFSA J. 2019, 17(1), e05539. doi: 10.2903/j.efsa.2019.5539.

I corrected the data according to the reviewer's comments.

Information from publications showing good citation (Xie, R.; Seip, H.M.; Wibetoe, G.; Nori, S.; McLeod, C.W. Heavy coal combustion as the dominant source of particulate pollution in Taiyuan, China, corroborated by high concentrations of arsenic and selenium in PM 10. Sci. Total Environ. 2006, 370, 409–415, doi:10.1016/j.scitotenv.2006.07.004):

Selenium concentrations were from 9.5 to 126 ng/m3, with an arithmetic average of 58 ng/m3. In general, the concentrations of urban particulate selenium are in the range of 1 to 10 ng/m3 (Ihnat et al., 1989). Extremely high concentrations of over 100 ng/m3 were reported in Ankara, Turkey (Ölmez and Aras, 1977). These samples were collected during the fall and winter in an area where peat was the major residence fuel and no emission control was in place.

2) lines 89-91 “The amount of selenium in groundwater (50–90 µg/L) is much higher than that in seawater (0.09 µg/L) [13] because of selenium elution from the parent rocks and excessive fertilization of soils with mixtures rich in selenium compounds [17].” 

Ref 13 1nd 17 deals with India data where high levels of selenium are rather typical. More common are concentrations less than 10 ug/L, usually 0.2-2 ug/L even in India (see “Speciation of selenium in groundwater: Seasonal variations and redox transformations”- Journal of Hazardous Materials Volume 192, Issue 1, 15 August 2011, Pages 263-269). Other countries including European ones demonstrate much lower values. 

For instance in Poland Se content in groundwater is 0.15 ug/l (Total content of arsenic, antimony and selenium in groundwater samples from western Poland//Pol. J. Environ. Stud. 2001;10(5):347–350

If the author desires not to change the above levels it is necessary to indicate the country where these levels were found. Strange that the Polish data are not included in this Polish review!!

I added a new citation and information about selenium content in Poznań (Poland).

Etim, E.U. Occurrence and distribution of arsenic, antimony and selenium in shallow groundwater systems of Ibadan metropolis, southwestern Nigerian. J. Health Pollut. 2017, 7(13), 32-41. doi:10.5696/2156-9614-7-13.32.

The selenium content in waters in countries is different. It all depends on fertilization, industry, pollution, etc.

3) line 185 “dimethyl selenite” should be changed to “dimethyl selenide” 

I changed the data

4)line 230 “glutathione peroxidase” – better write “glutathionperoxidases”, as there exists a family of 8 glutathione peroxidases in mammals (see BBA Vol 1830, Issue 5, May 2013, Pages 3289-3303 “Glutathione peroxidases”)

Of course, I agree with the reviewer. Glutathione Peroxidases are very much. But in the text is the general description of the enzyme as one.

5)line 290 “the main source of selenium is milk” should be changed to “cereals” first of all

I changed the data

6)line 291 the author did not change the words: the important source of selenium is “salt from salt mine”- if you leave these words unchanged it is necessary to indicate the country where this data was obtained 

I deleted this information from the manuscript

Similarly the sentence that “selenium is found in tomatoes, garlic, asparagus, kohlrabi etc” brings little information as all plants, animals, water, etc- contain this or that level of selenium (as also other elements). No changes according to this comment were made by the author. It is necessary to speak only about significant sources of selenium. Even garlic being a secondary accumulator of selenium contains low levels of selenium for instance in European countries. 2 

Of course, the individual vegetables contain selenium in their structure for which the contents of this element are given in the table.

I added information that these are vegetables enriched with this element.

Tomatoes do not belong to accumulators of selenium and usually they contain about 10 mcg Se/kg dry wheight!!! Calculate how much it will be per fresh weight knowing that the dry matter content in tomatoes is much lower than 10%)!!!! 

I deleted the information about tomatoes.

% water = 100% -% dry matter

% water = 100% - 10.00% = 90.00% - the percentage of water content

Relative error = (in theory - in practice) / (in theory) * 100%

The contents of the garter will be 9 times more

7) line 296. A large content of selenium can be found in plants (hyperaccumulators), e.g Astragalus bisulcatus and Brassica.” better write “ and several representatives of Brassicaceae genera” because not all Brassica species belong to selenium hyperaccumulators

I changed the information according to the reviewer's comments.

We hope that all responses to the Reviewers remarks would be comprehensive and sufficient and that the revised manuscript could be accepted for publication in Molecules. We are looking forward to receiving your final decision. 

Yours sincerely, 

Marek Kieliszek 

Faculty of Food Sciences, 

Department of Biotechnology, Microbiology and Food Evaluation, 

Warsaw University of Life Sciences, 

Nowoursynowska 159C, Warsaw, Poland